# Lower Prevalence of Chronic Pain in Manifest Huntington’s Disease: A Pilot Observational Study

**DOI:** 10.3390/brainsci12050676

**Published:** 2022-05-21

**Authors:** Marianna Delussi, Vittorio Sciruicchio, Paolo Taurisano, Francesca Morgante, Elena Salvatore, Isabella Pia Ferrara, Livio Clemente, Chiara Sorbera, Marina de Tommaso

**Affiliations:** 1AOU Policlinico, Applied Neurophysiology and Pain Unit, Basic Medical Sciences, Neurosciences and Sense Organs Department, Aldo Moro University, 70124 Bari, Italy; paolo.taurisano@uniba.it (P.T.); livio.clem@gmail.com (L.C.); marina.detommaso@uniba.it (M.d.T.); 2Children Epilepsy and EEG Center, PO, San Paolo ASL (Azienda Sanitaria Locale), 70019 Bari, Italy; vsciru@tin.it; 3Neurosciences Research Centre, Molecular and Clinical Sciences Research Institute, St George’s University of London, London SW17 0RE, UK; fmorgante@gmail.com; 4Department of Experimental and Clinical Medicine, University of Messina, 98951 Messina, Italy; 5AOU Federico II, Department of Advanced Biomedical Sciences, Università di Napoli, 80138 Napoli, Italy; e.salvatore@unina.it (E.S.); isabella.ferrara.pia@gmail.com (I.P.F.); 6IRCCS Centro Neurolesi Bonino Pulejo, 98124 Messina, Italy; chiarasorbera@gmail.com

**Keywords:** Huntington’s disease, premanifest subjects, manifest patients, family controls, chronic pain, pain features, motor impairment, cognitive decline, basal ganglia, pain interview

## Abstract

Pain is a minor problem compared with other Huntington Disease (HD) symptoms. Nevertheless, in HD it is poorly recognized and underestimated. So far, no study evaluated the presence of chronic pain in HD. The aim of this pilot study was to evaluate the presence and features of chronic pain in a cohort of HD gene carriers. An observational cross-sectional study was conducted in a cohort of HD gene carriers compared to not gene carriers (n.134 HD subjects, n.74 not gene mutation carriers). A specific pain interview, alongside a neurological, cognitive and behavioural examination, was performed in order to classify the type of pain, subjective intensity. A significant prevalence of “no Pain” in HD was found, which tended to increase with HD progression and a reduced frequency of pain in the last 3 months. A clear difference was found between manifest and premanifest HD in terms of intensity of pain, which did not change significantly with HD progression; however, a tendency emerges to a progressive reduction. No significant group difference was present in analgesic use, type and the site of pain. These findings could support a lower prevalence of *chronic pain* in manifest HD. Prevalence and intensity of *chronic pain* seem directly influenced by the process of neurodegeneration rather than by an incorrect cognitive and emotional functioning.

## 1. Introduction

Huntington’s disease (HD) is a progressive neurodegenerative illness with involuntary movements, cognitive decline, and varying degrees of behavioural and psychiatric dysfunction [1]. Neurodegenerative changes in HD primarily occur after a selective neuronal cell death and fibrillary astrocytosis in the caudate (medium spiny neurons of striatum), putamen, cerebral cortex, and to a lesser extent in the hippocampus and subthalamic nucleus, and in the cerebellum for juvenile cases [2,3]. In HD gene carriers, the increased CAG repeat that encodes the Huntingtin protein causes a progressive long polyglutamine repeat, resulting in a neural loss in the brain, particularly important in the basal ganglia. As a clinical consequence, motor, neurocognitive and neurobehavioral symptoms are observed in HD [4]. The basal ganglia have a key role in pain processing and analgesia; in fact, recent neuroimaging studies indicated a role of basal ganglia in the integration of motor, emotional, autonomic and cognitive responses to pain [5,6].

Pain may be a minor problem compared with other patient’s symptoms. Nevertheless, in HD it is poorly recognized and underestimated, even though it could play a key role in the quality of life of affected individuals. In Parkinson’s disease, chronic pain is a relevant non-motor symptom with an impact on health-related quality of life [7], for which an aberrant function in cortico-basal ganglia loops has been hypothesized [8]. There are several studies reporting possible causes of chronic pain in patients with HD. Mutant Huntingtin may contribute to muscle and endocrine dysfunction, potential causes of nociceptive and neuropathic pain in HD [9], possibly mediated by immunological and inflammatory process [10]. Dysfunction in cytokines and tryptophan kynurenines metabolism has been detected in neurodegenerative diseases, including HD [11], as well as in different chronic pain conditions [12]. Post mortem and animal studies, suggested the potential role of endocannabinoid system in the evolution of HD, though the relationship with possible chronic pain feature is still unclear [13].

Few and often conflicting studies have examined the prevalence and impact of pain in HD patients. Experimental studies based on pain showed a behavioural reaction to nociceptive stimulation not dissimilar from controls group [14]. In a previous study conducted with laser evoked potentials (LEPs), we found increased latencies of cortical responses, and supposed a possible disturbance in the central processing of pain stimuli with unclear clinical correlates [15]. A study evaluated pain severity in a cohort of 1474 people who participated in the European Huntington’s Disease Network (EHDN) REGISTRY study [15]. Pain severity was measured using one item of the Medical Outcome Study 36-item short form health survey which indicated a positive response in 41% of the sample, with higher scores in the more advanced stages of the disease. The authors thus underlined that the proportion of HD affected by pain is comparable to other neurodegenerative diseases [16]. A meta-analysis confirmed the overall mean prevalence of pain in HD to be around 41%, while the pain burden was found lower in HD compared to that in the general population [17]. The same authors conducted a cross-sectional analysis of the Enroll-HD study in premanifest, manifest HD gene mutation carriers (*n* = 3989 and *n* = 7.485, respectively) and in non-HD gene mutation carriers (*n* = 3719), using the same SF-36 subscore. They confirmed that the prevalence of pain interference was significantly higher in the middle stage of HD in respect to FC, with lower prevalence of painful conditions in the late and middle stage of HD [18]. So far, no study evaluated the presence of chronic pain in HD, using clinical examination and specific questionnaires for pain symptoms and disability. To this aim, in the present pilot study we have evaluated the presence and features of chronic pain in a cohort of HD gene carriers, compared to FC, with a specific interview and examination, useful to classify the type of pain, subjective intensity, and disability.

## 2. Materials and Methods

### 2.1. Study Design and Subject

This was a pilot observational cross-sectional study in a preliminary small sample; patients and their relatives were interviewed during the period between January 2019 and January 2021 at The Apulian Referral Center for HD, AOU Policlinico, Bari, Italy. Data on pain features were also collected at the IRCSS Bonino-Pulejo, Messina, Italy, and at Neurologia AOU Federico II, Napoli, Italy. Study groups inclusion criteria taken into account for the present study were: age ≥ 18, a positive genetic HD test, or first degree familiarity with HD gene carriers; exclusion criteria, for the clinical group as for the relatives, was a past or ongoing history of neurological and psychiatric conditions. We planned a 1:1 enrolment (1 HD carrier: 1 FC), but subjects recruitment was strongly limited during SARS-Cov-2 pandemic.

### 2.2. Outcomes

The clinical assessment included age, sex, HD status, CAG-repeat length, the Total Motor Score (TMS) as part of the Huntington’s Disease Rating Scale (UHDRS), the Total Functional Capacity Scale (TFC) [19], and the Mini-Mental State Examination (MMSE) [20]. HD subjects recruited at the Bari HD Center (n.68) were also evaluated with the PBA-s [21] for the psychiatric assessment and with Symbol Digit Modality Test (SDMT) [22], Categorical Verbal Fluency (FAS) [23], Stroop Test (ST) [24] for cognitive assessment. Details of the entire assessment (genetic investigation, neurological, psychiatric and cognitive) are previously described in Delussi et al. 2020 [25], the study sample were divided into 6 groups [26], as detailed in Table 1 and Table 2.

### 2.3. Pain Interview

We performed the following structured 5-point interview: Point A: “Have you felt pain during the last 3 months?”, dichotomous possibility of response, “Yes” or “No”. Point B: For how many days, during the last 3 months, have you felt pain?”, polytomous possibilities of response, “≤10”, “≤20”, “≤30”, “≤45”, “≤90”. Point C: “Where was your pain predominantly located?”, polytomous possibilities of response, “localized to Arms/Legs”, “Head”, “localized to Back or diffuse”. Point D: “How many days did you take pain medication during the last 3 months?”, polytomous possibilities of response, “≤10”, “≤20”, “≤30”, “≤45”, “≤90”. Point E: “How intense has your pain been during the past 3 months?” with 11-point numerical scale, from 0 to 10. All study subjects who reported “Yes” to Pain Interview Point A, were administered the BPI-s scale [27]. Based on clinical symptoms, clinical history, and neurological examination, pain was classified as nociceptive, nociplastic, neuropathic, and mixed; headaches were classified as nociplastic pain [28].

### 2.4. Genetic Investigation

The genetic test was performed on peripheral blood lymphocytes in order to define the condition of certain carrier by detecting the expansion of the CAG trait 40 in an allele of the IT-15 gene.

The Ethical Committee of Bari Policlinico General Hospital approved the study (The protocol number: 0038493/06/05/2019/AOUGP23/COMET/P), and each subject signed an informed consent.

### 2.5. Overall Data Socio-Demographic Characteristics

The whole study group included n.208 subjects (Table 1) The study sample was divided into different populations: Not Clinical group = Non-mutation carriers (FC) and Clinical group (HD) = Subjects with CAG > 39. The overall Clinical group = *n* 134 was divided into Pre-Manifest (HD-PM) and Manifest carriers (HD-M). HD-M was further divided into Manifest-Early (HD-M-Stage 1), Manifest-Middle (HD-M-Sage 2), and Manifest-late (HD-M-Stage 3). Demographic characteristics and all study outcomes are presented in Table 1 and Table 2.

### 2.6. Statistical Analysis

Preprocessing was performed in IBM SPSS Statistics Analysis Version 26 (IBM Corporation, Armonk, NY, USA). Two-sided 95% confidence interval (CIs) and *p* values ≤ 0.05 were considered statistically significant. A frequency analysis was performed to identify the presence of pain, the type of pain between study populations and the number of painful days during the last 3 months, considering clinical diagnosis and HD stage; the chi-square test was performed. A one-way ANOVA was performed, with Bonferroni post hoc test, in order to investigate how the pain intensity (VAS) changed among cases reporting pain in single groups. A two-way analysis of variance, with the post hoc Bonferroni test, was carried out between the BPI-I and BPI-F scores in the 3 different populations (FC, HD-M and HD-PM). In HD gene carriers, we performed a multiregression analysis considering the Vas values as dependent variable, and main clinical features, CAG, UHDRS, MMSE and TFC as predictors. In the subgroup of patients assessed at the Bari Center, the multiregression analysis evaluating the predictive factors for pain intensity in HD gene carriers, included the five PBA-s scores (Depression, Irritability, Psychoticism, Apathy, and Executive Function score) and Cognitive Variables (SDMT C-score and SDMT E-score, SCR E-score and SCR C-score, SCR E-score, SCN E-score and SCN C-score, SI E-score, SI C-score and SI Self C-score).

## 3. Results

We found a significant prevalence of “no *Pain”* responses in the HD-M group (chi-square Pearson test, 9.8 *p* = 0.007 (Table 2 and Figure 1a). Considering the stage of illness, “no *Pain”* tended to increase in stage 3 and 4, but the chi-square test was not significant (chi-square = 4.06; *p* = 0.31) (Figure 1b). The type of pain (neuropathic, nociceptive, putative and mixed pain) was similarly distributed among the different populations (FC, HD-M, and HD-PM) (chi-square = 6.59; *p* = 0.36) (Figure 2a). The number of chronic pain days during the last 3 months (0, ≤10, ≤20, ≤30, ≤45, ≤90) was different among the three populations (chi-square = 19.75; *p* = 0.014) (Figure 2b), as rarely recurrent pain for less than 10 days in the 3 months prevailed in HD-M and HD-PM. The use of symptomatic drugs was equally distributed among the groups (chi-square = 9.27; *p* = 0.32) (Figure 2d). The site of pain was similar among groups (Figure 2c). In the HD-PM group, headache slightly prevailed. Among subjects reporting pain in the arms/legs, it was generally of nociceptive type, a part from 1 FC and 1 HD-M patient with lumbar radiculopathy. Among patients with pain in the back, it was chronic low back pain, apart from two women with fibromyalgia in the FC group (Figure 2c). In the cases reporting pain, the subjective perception of intensity (0–10 VAS) was reduced in HD patients, compared to HD-PM group and FC (ANOVA with HD status as factor: F = 29.1; *p* < 0.001) (Figure 3a) Among HD gene carriers, VAS was different (F = 5.85; *p* = 0.002). The Bonferroni test indicated significantly higher Vas between HD-PM and HD-M subjects in middle and late stages. In Figure 3b, VAS values in the total of HD gene carriers are reported (Figure 3b). The BPI items (Figure 4a,b) were significantly different among FC, HD-M and HD-PM (Anova chi square = 79; *p* < 0.0001). The intra-subject comparison showed a significant difference in BPI-intensity sub-scores among groups (F = 24.98; *p* < 0.0001). The Bonferroni test between HD-M and other groups was significant (BPI I F *p* < 0.001). Taking into consideration the HD gene carriers, the ANOVA with HD stage as factor was significant (F = 5.97; *p* = 0.002). The intra-subject comparison was significant for both BPI frequency (*p* = 0.002) and intensity (*p* < 0.001). For BPI intensity, the Bonferroni test between PM and early (*p* < 0.05), middle and late stage (*p* < 0.001) HD patients was significant. For BPI frequency, the Bonferroni test was not significant. In the HD gene carriers, the multiregression analysis with VAS as dependent variable and TFC, CAG triplets number UHDRS and MMSE as predictors, was significant (r-square = 0.31; ANOVA F = 9.7; *p* < 0.0001). In particular, higher TFC scores and lower CAG scores, predicted higher VAS values (TFC β = 0.37; t = 2.26; *p* = 0.027; CAG β = −0.21; t = −2.02; *p* = 0.047) (Figure 5a,b). There was an association between Vas scores and overall cognitive variables taken together (SDMT C-score and SDMT E-score, SCR E-score and SCR C-score, SCR E-score, SCN E-score and SCN C-score, SI E-score, SI C-score and SI Self C-score r square = 0.30 F 2 *p* = 0.044), though no single test reached the statistical significance. No correlation was found between VAS score and PBA items (r square = 0.1; F = 1.34; *p* = 0.25).

## 4. Discussion

The prevalence of pain in the overall study sample (n 208) is precisely split in half (Y104/N104, Y50% vs. N50%), with reduced prevalence of pain in manifest HD patients. The type of pain was similar among groups, but manifest HD had a reduced frequency of pain in the last 3 months. A clear difference emerged between manifest and premanifest HD in terms of intensity of pain and related disability, which did not change significantly with HD progression, though there was a tendency to a progressive reduction. In the group of HD gene carriers, the intensity of pain decreased with decrease in TFC and was higher in subjects with higher CAG expansion. These data are only in partial agreement with previous studies on prevalence of pain in HD samples, all performed using the sub-items of SF-36 included in Registry and Enroll-HD database. Underwood et al. [16] found that the prevalence of pain in the total HD sample was 41%, increasing with disease progression. Pain severity was significantly associated with participant-rated anxiety and depression. One study showed an increase in the prevalence of pain in HD from 32% in the premanifest stage to 50% in the late stage [29]. A recent meta-analysis estimated the overall mean prevalence of pain in HD to be around 41% (95% confidence interval: 36–46%) [30]. The same authors conducted a further analysis on a large HD sample, included in the fourth periodic database of the Enroll-HD study, using the same questionnaire. Results showed that the pain burden, measured in terms of pain intensity and interference, was lower in HD compared to that in the general population [30]. Our pilot study was conducted with a different method, based on the direct interview and examination of patients and family controls, aiming to put a specific diagnosis of chronic pain, according to current diagnostic criteria [31] and measures of pain intensity and disability as numerical VAS and BPI. We evaluated the presence of pain in a longer period (3 months), according to chronic pain definition [31], instead of asking for presence of pain in the last week. This approach does not enable a direct comparison with prevalence obtained in previous studies, but could shed light on pain as a symptom not to be ignored. We found that in the majority of cases, pain recurrence was rare, less than 10 days in the last 3 months, especially for subjects reporting headache. However, it affected 50% of family controls, which is in line with the global prevalence of chronic pain [32,33]. The type of pain was in prevalence nociceptive and nociplastic, the latter including headaches, in keeping with epidemiology of chronic pain in the general population [33]. In the HD patients group, the representation of chronic pain subtypes was similar to family controls, while nociplastic pain, which included in large part headaches, showed a tendency to over expression in premanifest HD. This observation is worth to be confirmed in larger population study. Our preliminary results indicate that chronic pain is less frequent in manifest HD, with a tendency to reduce over disease progression. Pain intensity and related disability were also milder in HD patients. Moreover, this phenomenon was absent in the premanifest phase, and became relevant in the middle and late stage of the disease, with the full phenotypical expression of the disease. Reduced pain perception seemed to characterize patients with larger CAG expansion and reduced functional capacity, in accord with the results of multiregression analysis. Moreover, intensity perception and related disability, had a weak correlation with cognitive impairment, and no correlation at all with psychiatric features as measured with the PBA-s, probably due to the small sample size. As a matter of fact, reduced pain perception could be an intrinsic phenotypical expression of the disease, just related to the primary degenerative process involving the basal ganglia. The basal ganglia have a key role in pain processing and analgesia; indeed, recently, neuroimaging studies has added important information on their activation in conditions of acute pain and chronic pain. Alterations in cortical and sub-cortical regions in pain suggests that the basal ganglia are uniquely involved in thalamo-cortico-basal ganglia loops to integrate many aspects of pain [8]. These include the integration of motor, emotional, autonomic and cognitive responses to pain [5]. On the other hand, there is conjoining experimental and clinical evidence supporting a fundamental role of the basal ganglia as a sensory analyser engaged in central somatosensory control [34]. Animal studies suggested that mutant HTT leads to altered pain behaviour and pain-related cytokine response [35]. In previous studies, our group demonstrated that impaired basal ganglia function in HD causes an alteration in pain processing with a significant prolongation of processing painful stimuli, even in pre-manifest stage of HD [36]. Moreover, we also observed an increased threshold of RIII response in HD patients, a neurophysiological signature of reduced pain perception [37]. The basal-ganglia show some of the earliest changes in HD-PM, with loss of striatal grey matter grey matter [38] and white matter [39] occurring 15 years before disease onset. Nevertheless, chronic pain prevalence and pain perception were similar between HD-PM and controls. These data request confirmation in larger series, but suggest that the abnormalities in pain processing could not be attributable to the early phase of the disease, but become manifest in the course of HD progression. Regarding this point, our results are in accord with previous studies in large HD samples, showing a decline in pain burden in the advanced stage [40].

### Study Limitations

This was a pilot study aiming to a specific evaluation to support a diagnosis of chronic pain in HD cohorts. Number of cases are small, and all the reported data need to be confirmed in larger groups. The unbalance of sample size between PM-HD, M-HD and FC groups could have a negative impact on statistical analysis. Moreover, this limitation was partly due to the pandemic restriction. 

## 5. Conclusions

These preliminary data could support a lower prevalence of chronic pain in manifest HD. The prevalence and intensity of pain seem directly influenced by the process of neurodegeneration rather than by an incorrect cognitive and emotional functioning. In fact, the thalamo-cortico-basal ganglia loops engaged in central pain control could go through a progressive dysfunction in the course of the disease. Other neurodegenerative diseases as dementia, motoneuronal and extrapyramidal disorders are opening a new scenario on pain syndromes, which represent an underestimated problem [39]. The improvement of pain symptoms evaluation could be important in clinical management of patients. The confirmation of a lower pain sensitivity in advanced HD status, could enhance the attention of care givers toward possible silent causes of inflammation [41] or neuropathic conditions. 

The present interview was conducted during the routine clinical examination, being quite simple and not expensive in terms of time. In the later stage of the disease, the application of specific pain scales for severe cognitive decline could also be suitable [14,42]. Considering the above, it is increasingly appropriate to include specific evaluation of chronic pain in the large multicentre databases, in order to understand the real impact of chronic pain in the global burden of the disease.

## Figures and Tables

**Figure 1 brainsci-12-00676-f001:**
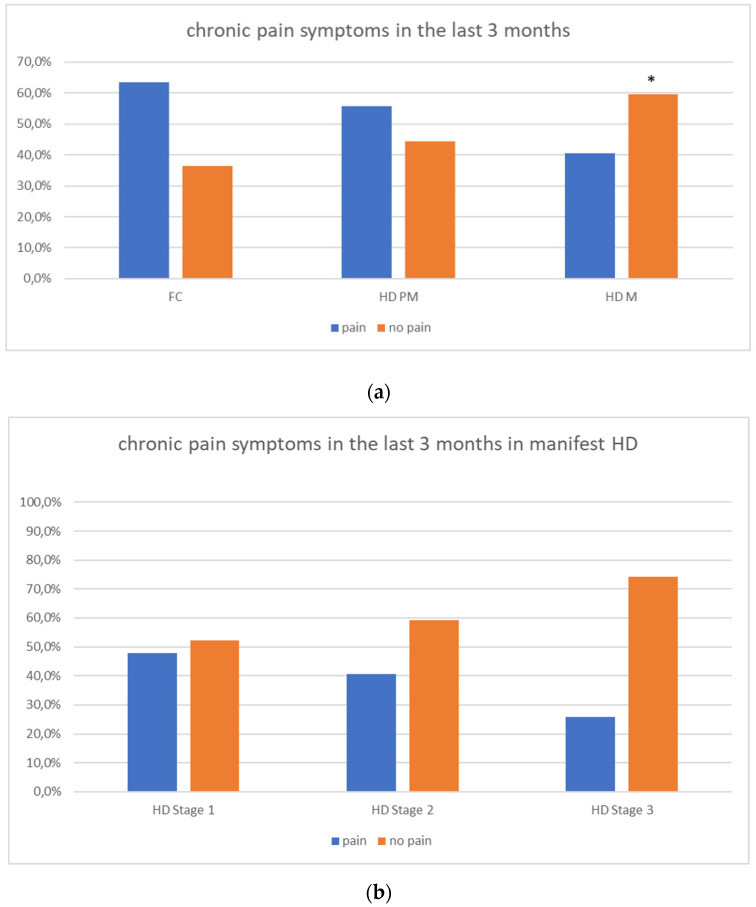
(**a**): prevalence of no pain responses in HD groups. (**b**): prevalence of no pain responses in HD stages. “*” = statistically significant value.

**Figure 2 brainsci-12-00676-f002:**
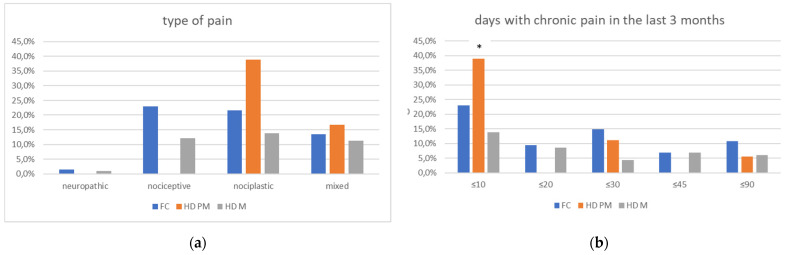
(**a**): prevalence of no pain responses among type of pain among FC, HD-M and HD-PM groups. (**b**): % of pain days during the last 3 months among FC, HD-M and HD-PM groups. (**c**): site of pain was similar among FC, HD-M and HD-PM groups: Blue: FC, Orange: HD-PM, Grey: HD-M. (**d**): use of symptomatic drugs among FC, HD-M and HD-PM groups: Blue: FC, Orange: FC, Grey: HD-M. “*” = statistically significant value.

**Figure 3 brainsci-12-00676-f003:**
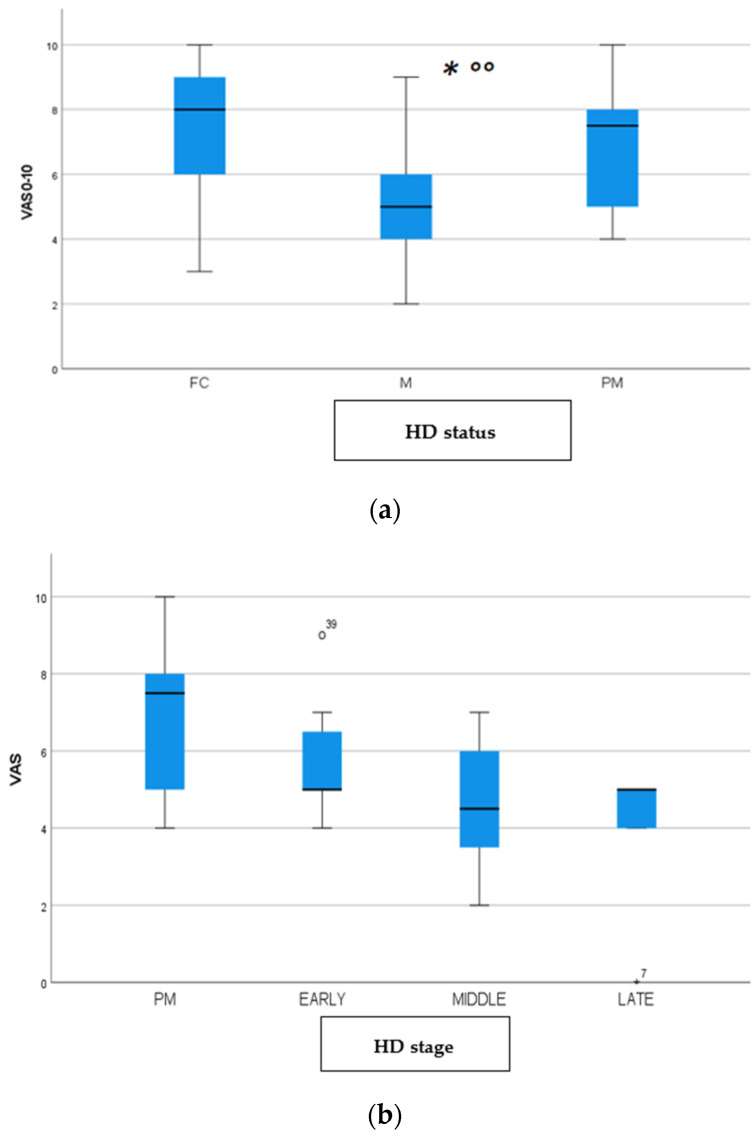
(**a**): Intensity of pain (VAS) among FC, HD-M and HD-PM groups. (**b**): Intensity of pain (Vas) among HD gene carriers groups. “*” = statistically significant value.

**Figure 4 brainsci-12-00676-f004:**
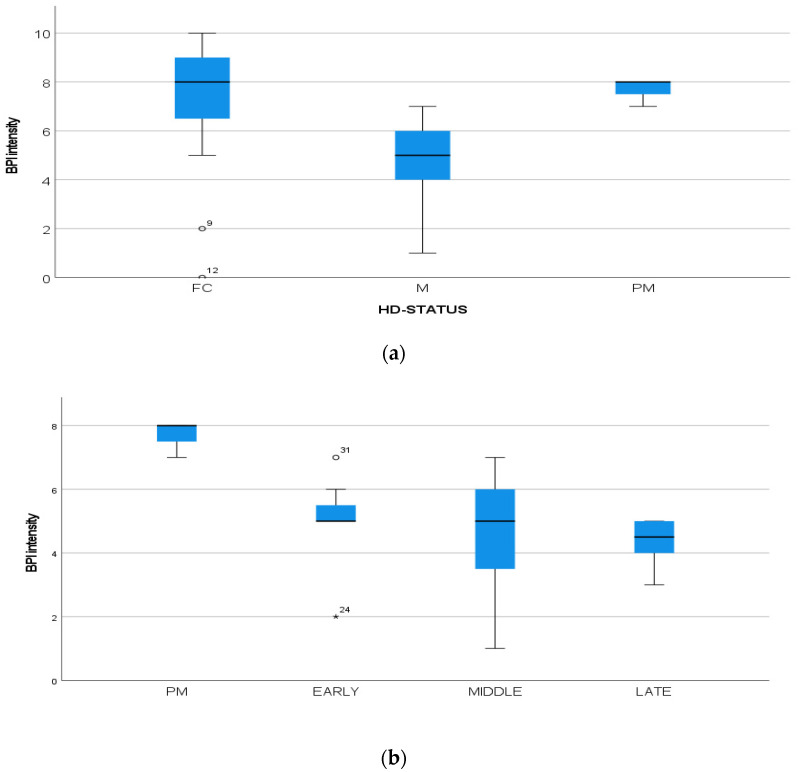
(**a**): Intensity of pain (BPI) among FC, HD-M and HD-PM groups. (**b**): Intensity of pain (BPI) among HD gene carriers groups. “*” = statistically significant value.

**Figure 5 brainsci-12-00676-f005:**
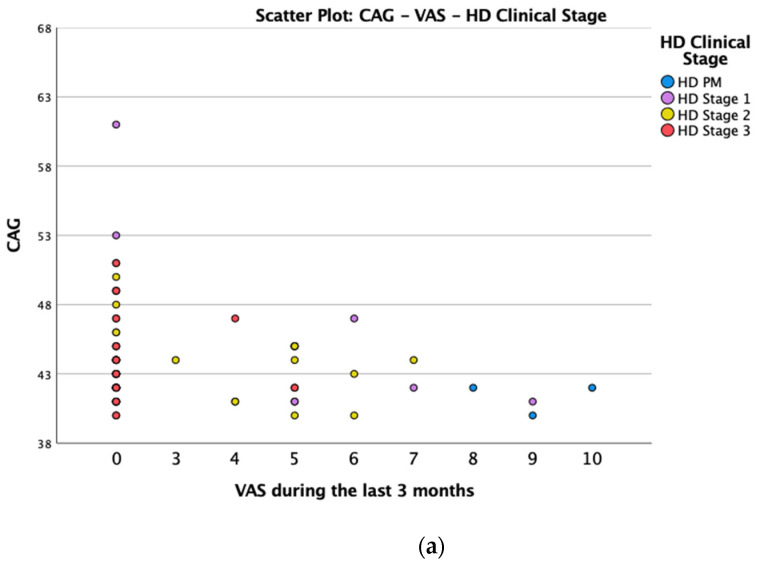
(**a**): Scatter plot CAG and VAS between HD gene carriers groups. (**b**): Scatter plot TFC and VAS between HD stages.

**Table 1 brainsci-12-00676-t001:** Demography and Outcome of the entire study group.

	Overall Sample	Non Mut. Carriers	Pre HD	M Tot	HD Early	HD Middle	HD Late
*n*	208	74	18	116	23	59	27
Age Mean/DS	53.59/11.88	55.19/10.52	45.56/10.38	53.88/12.46	51.57/13.56	54.8/9.38	55.3/17.42
Gender N	110F/98M	43F/31M	7F/11M	60F/56M	13F/10M	27F/32M	16F/11M
%	52.9/47.1	58.1/41.9	38.9/61.1	51.7/48.3	56.5/43.5	45.8/54.2	59.3/40.7
CAG Mean/DS	-	-	41.25/0.95	44.15/3.65	45.36/6.36	43.48/2.55	44/3.26
Y. of Illness Mean/DS	-	-	-	11.3/7.22	8.36/8.65	10.29/5.71	13.41//7.99
UHDRS Mean/DS	-	-	-	46.83/23.53	21.69/14.59	44.1/19.93	64.88/18.14
TFC Mean/DS	8.67/4.62	12.62/1.18	13/0	6.01/4.12	12.43/0.78	5.46/2.01	1.74/2.19
MMSE Mean/DS	24.34/6.43	28.81/2.02	29.14/0.86	20.61/6.47	27.3/2.51	19.96/4.63	13.76/6.25
SDMT Corr. Mean/DS	11.91/16.75	33.53/13.05	20.75/17.65	6.11/12.62	29.2/14.62	2.24/5.85	0.96/4.58
SDMT Err. Mean/DS	0.09/0.32	0	0.5/1	0.08/0.27	0.2/0.42	0.07/0.25	0.04/0.2
SCN Corr. Mean/DS	33.95/30.24	70.47/15.54	54.25/15.01	21.45/23.99	58.1/22.49	22.83/16.94	3.78/9.23
SCN Err. Mean/DS	0.08/0.35	0	0.25/0.5	0.1/0.39	0/0	0.17/0.53	0.04/0.2
SCR Corr. Mean/DS	40.92/35.6	83.21/13.1	70.5/27.5	26.05/28.61	69.8/25.58	27.62/19.74	5.04/13.09
SCR Err. Mean/DS	0.07/0.33	0	0	0.1/0.39	0/0	0.07/0.37	0.17/0.49
SI. Corr. Mean/DS	17.42/19.38	43.47/9.73	39.25/11.14	8.03/12.3	27.4/15.1	6.9/8.44	1.04/3.47
SI. Err. Mean/DS	0.6/1.38	0.11/0.31	0.5/0.57	0.76/1.57	0.6/0.84	1.41/2.06	0/0
PBA-s Dep. Mean/DS	4.11/3.47	1.5/2.43	2.5/2.38	4.97/3.4	1.7/2.16	5.07/3.52	6.26/2.8
PBA-s Irr. Mean/DS	2.7/3.72	1.11/2.34	3/3.83	3.15/3.95	1.2/2.44	4.03/4.75	2.87/3.06
PBA-s Psyc. Mean/DS	0.67/2.97	0	0	0.9/3.43	0/0	0.21/0.77	2.17/5.41
PBA-sApat. Mean/DS	1.68/3.18	0	0.5/1	2.24/3.53	0.1/0.3	1.86/2.56	3.65/4.68
PBA-sEx.Fu. Mean/DS	1.64/4.2	0	0	2.23/4.76	0.1/0.31	2.17/3.45	3.22/6.68

**Table 2 brainsci-12-00676-t002:** Pain Interview and BPI-s outcomes.

	Overall S	Non Mut. C.	Pre HD	M Tot	HD Early	HD Middle	HD Late
Point A							
YES %	104/50	47/63.5	10/55.6	47/40.5	11/47.8	24/40.7	7/25.9
No %	104/50	27/36.5	8/44.4	69/59.5	12/52.2	35/59.3	20/74.1
Point							
≤10 N/%	40/19.2	17/23	7/38.9	16/13.8	1/4.3	8/13.6	5/18.5
≤20 N/%	17/8.2	7/9.5	0/0	10/8.6	3/13	6/10.2	1/3.7
≤30 N/%	18/8.7	11/14.9	2/11.1	5/4.3	0/0	3/5.1	1/3.7
≤45 N/%	13/6.3	5/6.8	0/0	8/6.9	3/13	4/6.8	0/0
≤90 N/%	16/7.7	8/10.8	1/5.6	7/6	2/8.7	3/5.1	0/0
Point C							
Ar.-Le. N/%	40/19.2	14/18.9	1/5.6	25/21.6	6/26.1	11/18.6	5/18.5
He. N/%	38/18.3	20/27	6/33.3	12/10.3	3/13	7/11.9	0/0
Ba.-Sh. N/%	22/10.6	12/16.2	2/11.1	8/6.9	2/8.7	4/6.8	2/7.4
Tr. N/%	3/1.4	0	1/5.6	2/1.7	0/0	2/3.4	0/0
Point D							
≤10 N/%	70/33.7	24/32.4	9/50	37/31.9	7/30.4	20/33.9	7/25.9
≤20 N/%	15/7.2	11/14.9	0/0	4/3.4	0/0	3/5.1	0/0
≤30 N/%	8/3.8	7/9.5	1/5.6	0/0	0/0	0/0	0/0
≤45 N/%	3/1.4	3/4.1	0/0	0/0	0/0	0/0	0/0
≤90 N/%	6/2.9	2/2.7	0/0	4/3.4	2/8.7	1/1.7	0/0
Point E							
VAS							
Mean/DS	6.32/2.23	7.67/1.93	6.9/2.02	4.85/1.57	4.82/1.94	4.63/1.4	4.83/0.4
BPI-s I Mean/DS	5.87/2.68	7.09/2.73	7.71/0.48	4.3/1.87	4/2.14	4.43/1.91	4.43/0.78
BPI-s F Mean/DS	5.3/3.19	5.94/1.73	4.81/0.93	4.67/4.41	8.87/8.18	3.88/2.36	3.27/1.91
Pain type							
Neuropathic							
N/%	2/1	1/1.4	0/0	1/0.9	0/0	1/1.7	0/0
Nociceptive							
N/%	31/14.9	17/23	0/0	14/12.1	2/8.7	5/8.5	5/18.5
Nociplastic							
N/%	39/18.8	16/21.6	7/38.9	16/13.8	3/13	12/20.3	0/0
Visceral							
N/%	1/0.5	1/1.4	0/0	0/0	0/0	0/0	0/0
Mixed							
N/%	26/12.5	10/13.5	3/16.7	13/11.2	5/21.7	5/8.5	1/3.7

## Data Availability

The data shared are in accordance with consent provided by participants on the use of confidential data.

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
