# Peer review of "Lower Prevalence of Chronic Pain in Manifest Huntington’s Disease: A Pilot Observational Study"

_brainsci, 2022, doi:10.3390/brainsci12050676_

Round 1

Reviewer 1 Report

20 April 2022

Manuscript ID: brainsci-1730462

Type: Article

Title: ‘Chronic Pain in Huntington’s Disease: a pilot observational Study’ by Delussi M et al., submitted to Brain Sciences

Dear Authors,

Huntington' s disease HD is a hereditary neurological disease characterized with uncontrolled movement of the arms, legs, head, face and upper body and cognitive impairment. Little is known about the prevalence of chronic pain in patients with HD. In the present review entitled ‘Chronic Pain in Huntington’s Disease: a pilot observational Study’, Delussi and colleagues conducted observational cross-sectional study to evaluate the presence and features of chronic pain in HD gene carriers. The main strength of this original article is that it addresses an interesting and timely question, revealing a significant prevalence of no pain which tends to increase with HD progression and a clear difference in pain intensity between premanifest and manifest HD. The authors concluded that the prevalence and intensity of chronic pain seem directly influenced by the process of neurodegeneration rather than by cognitive and emotional dysfunction.

In general, I think the idea of this article is really interesting and the authors’ fascinating observations on this timely topic may be of interest to the readers of Biomedicines. However, some comments, as well as some crucial evidence that should be included to support the author’s argumentation, needed to be addressed to improve the quality of the manuscript, its adequacy, and its readability before its publication in the present form. My overall judgment is to publish this article after the authors has carefully considered my suggestions below, reshaping parts of the abstract and the body of manuscript by reorganizing the introduction to orderly present concepts in general, rationales, and the purpose of this study. In general, I recommend the authors to use more evidence to back their claims, especially in the in introduction, the discussion, and the conclusion, which I believe are currently insufficient. Thus, I advise the authors to attempt to deepen the subject of their manuscript and to focus their efforts on researching relevant literature: I believe that adding more studies will help to provide better and more accurate background to this paper. In this study, I will try to help the author by presenting comments below to improve their manuscript.

Please consider the following comments:

  1. Title: I recommend presenting an appropriate title reflecting this manuscript.
  2. Abstract: According to the Journal’s guidelines, the abstract should be introduced as a single paragraph, and should follow the style of structured abstracts, but without headings. Please correct the actual one. I also advise the authors to abridge it to 200 words and make sure the elements of Abstract for original articles such as background, rationale, purpose, methods, results, and conclusion. Also, the authors need to present the descriptions of groups tested in this study clearly.
  3. A graphical abstract summarizing the manuscript is highly recommended.
  4. Keywords: I recommend listing ten keywords.
  5. Introduction: The ‘Introduction’ section is well-written and nicely presented, with a good balance of information about HD. Nevertheless, I believe that more information about the comorbidity of pain sensation in neurologic and psychiatric diseases in general, not limited to Parkinson’s disease, the mechanisms of pain sensation from the neuroanatomical and psychological points of view including inflammation, and known findings of the prevalence and features of pain sensation in HD, leading to the rationale, purpose, and potential of this study (https://doi.org/10.3390/ijms22041561; https://doi.org/10.1007/s40265-018-0996-1; https://doi.org/10.3390/biomedicines10040877; https://doi.org/10.3390/biomedicines9080897; https://doi.org/10.3390/ijms222011055; doi: 10.3390/ijms21176045; https://doi.org/10.3390/ijms21072431; https://doi.org/10.1213/ANE.0000000000002419).
  6. Methods and Materials: ‘2.2 Outcomes’ is probably a right choice of words for this subheading.
  7. Discussion: In this final section, the authors are advised to describe any practical or operational issues involved in performing the study; however, I would have liked to see some views on a way forward. Hence, I ask them to include some thoughtful as well as in-depth considerations, trying to explain the theoretical and practical application of their research.
  8. Discussion: Following the previous point raised, I would ask the authors discussing, by the end of the manuscript, the possible various drawbacks that this study protocol might have, expanding the limitation, in order to describe in detail and report all potential technical issues which might occur. Also it deserves to present the potential of this study, emphasizing in advances in understanding pain sensation in HD, the ultimate goal, knowledge and research to achieve this goal, and future research direction.
  9. References: The number of references it is dramatically low for original reviews, and this prevents the possibility of publishing it in this form. References should be more than 60-70 for original research articles. Authors should consider revising the bibliography, as there are several incorrect citations. Indeed, according to the Journal’s guidelines, they should provide the abbreviated journal name in italics, the year of publication in bold, the volume number in italics for all the references, and DOI numbers.

Overall, the manuscript contains five figures, two tables, and 35 references. I believe that this manuscript may carry important value evaluating the presence and features of chronic pain in HD gene carriers and HD patients. I hope that, after these careful revisions, the manuscript can meet the Journal’s standards for publication. I am available for a new round of revision of this protocol manuscript.

Best regards,

Reviewer

Author Response

Reviewer 1 Dear Authors,

Huntington' s disease HD is a hereditary neurological disease characterized with uncontrolled movement of the arms, legs, head, face and upper body and cognitive impairment. Little is known about the prevalence of chronic pain in patients with HD. In the present review entitled ‘Chronic Pain in Huntington’s Disease: a pilot observational Study’, Delussi and colleagues conducted observational cross-sectional study to evaluate the presence and features of chronic pain in HD gene carriers. The main strength of this original article is that it addresses an interesting and timely question, revealing a significant prevalence of no pain which tends to increase with HD progression and a clear difference in pain intensity between premanifest and manifest HD. The authors concluded that the prevalence and intensity of chronic pain seem directly influenced by the process of neurodegeneration rather than by cognitive and emotional dysfunction.

Thank you for your favourable impression of our manuscript. However, I wonder if there has been a misunderstanding about the type of manuscript, which was an original study and not a revision. We did, however, appreciate the advice, and implemented the bibliography by building on the recommended papers.

In general, I think the idea of this article is really interesting and the authors’ fascinating observations on this timely topic may be of interest to the readers of Biomedicines. However, some comments, as well as some crucial evidence that should be included to support the author’s argumentation, needed to be addressed to improve the quality of the manuscript, its adequacy, and its readability before its publication in the present form. My overall judgment is to publish this article after the authors has carefully considered my suggestions below, reshaping parts of the abstract and the body of manuscript by reorganizing the introduction to orderly present concepts in general, rationales, and the purpose of this study. In general, I recommend the authors to use more evidence to back their claims, especially in the in introduction, the discussion, and the conclusion, which I believe are currently insufficient. Thus, I advise the authors to attempt to deepen the subject of their manuscript and to focus their efforts on researching relevant literature: I believe that adding more studies will help to provide better and more accurate background to this paper. In this study, I will try to help the author by presenting comments below to improve their manuscript.

Please consider the following comments:

We have changed the title, now it sounds more accurate about our conclusions

  1. Title: I recommend presenting an appropriate title reflecting this manuscript.

We changed the title as

“Lower prevalence of chronic pain in manifest Huntington’s Disease: a pilot observational Study.

This title outlined main results

  1. Abstract: According to the Journal’s guidelines, the abstract should be introduced as a single paragraph, and should follow the style of structured abstracts, but without headings. Please correct the actual one. I also advise the authors to abridge it to 200 words and make sure the elements of Abstract for original articles such as background, rationale, purpose, methods, results, and conclusion. Also, the authors need to present the descriptions of groups tested in this study clearly.

Thank you for the observation. We have drastically reduced the number of words and corrected the text according to your suggestions

  1. Keywords: I recommend listing ten keywords.

We integrated the number of keywords

  1. Introduction: The ‘Introduction’ section is well-written and nicely presented, with a good balance of information about HD. Nevertheless, I believe that more information about the comorbidity of pain sensation in neurologic and psychiatric diseases in general, not limited to Parkinson’s disease, the mechanisms of pain sensation from the neuroanatomical and psychological points of view including inflammation, and known findings of the prevalence and features of pain sensation in HD, leading to the rationale, purpose, and potential of this study (https://doi.org/10.3390/ijms22041561; https://doi.org/10.1007/s40265-018-0996-1; https://doi.org/10.3390/biomedicines10040877; https://doi.org/10.3390/biomedicines9080897; https://doi.org/10.3390/ijms222011055; doi: 10.3390/ijms21176045; https://doi.org/10.3390/ijms21072431; https://doi.org/10.1213/ANE.0000000000002419).

Many thanks for your precious suggestions. We integrated the introduction with these important references. The last one was cited in the discussion

  1. Methods and Materials: ‘2.2 Outcomes’ is probably a right choice of words for this subheading.

Ok, done!

  1. Discussion: In this final section, the authors are advised to describe any practical or operational issues involved in performing the study; however, I would have liked to see some views on a way forward. Hence, I ask them to include some thoughtful as well as in-depth considerations, trying to explain the theoretical and practical application of their research.

  1. Discussion: Following the previous point raised, I would ask the authors discussing, by the end of the manuscript, the possible various drawbacks that this study protocol might have, expanding the limitation, in order to describe in detail and report all potential technical issues which might occur. Also it deserves to present the potential of this study, emphasizing in advances in understanding pain sensation in HD, the ultimate goal, knowledge and research to achieve this goal, and future research direction.

Thank you again for your comments. In reality, the pain interview we carried out was very simple, so we did not find any major drawbacks. It was simply included in the regular clinical visits.

Regarding "views on a way forward", we suggested that it could be a useful tool in clinical practice, together with a specific assessment for advanced cases with cognitive impairment, obviously after validation with stronger case studies.

  1. References: The number of references it is dramatically low for original reviews, and this prevents the possibility of publishing it in this form. References should be more than 60-70 for original research articles. Authors should consider revising the bibliography, as there are several incorrect citations. Indeed, according to the Journal’s guidelines, they should provide the abbreviated journal name in italics, the year of publication in bold, the volume number in italics for all the references, and DOI numbers.

We are sorry if the bibliography seemed inadequate, but in fact it was carefully selected. We have nevertheless appreciated the advice and implemented the bibliography.

This is not the first time we have submitted a paper to Brain Sciences and the number of references does not seem mandatory, especially for a pilot study on a little-studied topic such as pain in HD, a rare disease. The reviewer suggested relevant and interesting references that we have added to the original list. The list of references is now much more complete. We thank you for your comments, which may add value to our study. The requested changes are highlighted in yellow.

The Doi does not appear in the journal's guidelines for bibliography formatting; however, it has been added to all bibliographic entries.

The bibliography has all been carefully revised and the style formatted.

English has also been revised

Best regards,

All the authors

Reviewer 2 Report

I thank the authors for their study, supporting my clinical impression.  From my clinical impression pain and also temperature perception seems to be impaired in manifest HD, starting in earlier or middle stages. Patients have had a fall with broken rips and don’t complain. Some had cholecystitis and don’t complain. Thus, as a consequence, in my view, if a patient complains pain one need to be attentive and look for a possible reason for pain. This might be discussed. E.g.:

Upper gastrointestinal findings in Huntington's disease: patients suffer but do not complain. Andrich JE, Wobben M, Klotz P, Goetze O, Saft C.J Neural Transm (Vienna). 2009 Dec;116(12):1607-11.

On the other hand there are a very few patients with depression and severe somatoform depression complaining pain. Authors did not found a correlation to depression, thus perhaps this smaller collective there was none such patient, which also might be discussed.

 Involvement of Thalamo-Cortico-Basal Ganglia Loops as discussed and alteration of sensory – auditory - processing was discussed in an fMRI study too, supporting this approach.

fMRI reveals altered auditory processing in manifest and premanifest Huntington's disease. Saft C, Schüttke A, Beste C, Andrich J, Heindel W, Pfleiderer B.Neuropsychologia. 2008 Apr;46(5):1279-89. 

I only have a few minor comments:

Please define and cite the criteria for grouping into stage 1-3. It is UHDRS Stage 1-3 and no participants with Stage 4 and 5??

Perhaps harmonize the abbreviations, e.g. HD-M and in table M-tot, HD-M-Stage 1 and in table HD middle. I would suggest to use the same abbreviations throughout the text.

Figure 3 b is Italian

Tables are a lot and a bit confusing. Since type of pain was the same, I wonder if figure 2 is needed.

Author Response

Reviewer 2

I thank the authors for their study, supporting my clinical impression.  From my clinical impression pain and also temperature perception seems to be impaired in manifest HD, starting in earlier or middle stages. Patients have had a fall with broken rips and don’t complain. Some had cholecystitis and don’t complain. Thus, as a consequence, in my view, if a patient complains pain one need to be attentive and look for a possible reason for pain. This might be discussed. E.g.:

Upper gastrointestinal findings in Huntington's disease: patients suffer but do not complain. Andrich JE, Wobben M, Klotz P, Goetze O, Saft C.J Neural Transm (Vienna). 2009 Dec;116(12):1607-11.

Dear colleague,

it is a pleasure to know that our work reflects your clinical impressions. Thank you for your interesting remarks, which we have taken on board.

We have included a reflection based on your suggestion in the discussions. it seemed very appropriate to us.

On the other hand there are a very few patients with depression and severe somatoform depression complaining pain. Authors did not found a correlation to depression, thus perhaps this smaller collective there was none such patient, which also might be discussed.

Thank you for the suggestion, it was a fault. I have included a reflection on the PBA-s scores in the discussion.

 Involvement of Thalamo-Cortico-Basal Ganglia Loops as discussed and alteration of sensory – auditory - processing was discussed in an fMRI study too, supporting this approach.

thank you for your observation and supporting our inferences

fMRI reveals altered auditory processing in manifest and premanifest Huntington's disease. Saft C, Schüttke A, Beste C, Andrich J, Heindel W, Pfleiderer B.Neuropsychologia. 2008 Apr;46(5):1279-89. 

I only have a few minor comments:

Please define and cite the criteria for grouping into stage 1-3. It is UHDRS Stage 1-3 and no participants with Stage 4 and 5??

Perhaps harmonize the abbreviations, e.g. HD-M and in table M-tot, HD-M-Stage 1 and in table HD middle. I would suggest to use the same abbreviations throughout the text.

Figure 3 b is Italian

Tables are a lot and a bit confusing. Since type of pain was the same, I wonder if figure 2 is needed.

Thank you again for your attention to our work.

We have updated the manuscript according to your valuable considerations and corrections.

English has also been revised

Best regards,

All the authors

Round 2

Reviewer 1 Report

14 May 2022

Manuscript ID: brainsci-1730462

Type: Article

Title: ‘Lower prevalence of chronic pain in manifest Huntington’s Disease: a pilot observational Study’ by Delussi M et al., submitted to Brain Sciences

Dear Authors,

The authors did an excellent work clarifying the questions I have raised in the previous round of review. Currently, this paper is a well-written, timely piece of research and provides a useful study addressing an interesting and innovative question, evaluating the presence and features of chronic pain in HD gene carriers and patients with Huntington’s disease. Overall, this is a timely and needed work, thus I believe that manuscript now meets the Journal’s standards for publication. I am always available for other reviews of such interesting and important articles. I look forward to seeing further study on this issue by these authors in the future.

Thank You for your work.

Best regards,

Reviewer